# IMPROVING KNOWLEDGE DISTILLATION VIA REGULARIZING FEATURE DIRECTION AND NORM

## ABSTRACT

Knowledge distillation (KD) exploits a large well-trained `teacher` neural network to train a small `student` network on the same dataset for the same task. Treating `teacher`'s feature as knowledge, prevailing methods train `student` by aligning its features with the `teacher`'s, e.g., by minimizing the KL-divergence between their logits or L2 distance between their features at intermediate layers. While it is natural to assume that better feature alignment helps distill `teacher`'s knowledge, simply forcing this alignment does not directly contribute to the `student`'s performance, e.g., classification accuracy. For example, minimizing the L2 distance between the penultimate-layer features (used to compute logits for classification) does not necessarily help learn a better `student`-classifier. Therefore, we are motivated to regularize `student` features at the penultimate layer using `teacher` towards training a better `student` classifier. Specifically, we present a rather simple method that uses `teacher`'s class-mean features to align `student` features w.r.t their *direction*. Experiments show that this significantly improves KD performance. Moreover, we empirically find that `student` produces features that have notably smaller norms than `teacher`'s, motivating us to regularize `student` to produce large-norm features. Experiments show that doing so also yields better performance. Finally, we present a simple loss as our main technical contribution that regularizes `student` by simultaneously (1) aligning the *direction* of its features with the `teacher` class-mean feature, and (2) encouraging it to produce large-*norm* features. Experiments on standard benchmarks demonstrate that adopting our technique remarkably improves existing KD methods, achieving the state-of-the-art KD performance through the lens of image classification (on ImageNet and CIFAR100 datasets) and object detection (on the COCO dataset).

## 1 INTRODUCTION

Knowledge distillation (KD) is a specific type of methodology in model compression that aims to train a smaller model (called `student`) by distilling knowledge, i.e., what has been learned, in a larger `teacher` model (Hinton et al., 2015). Deploying the small `student` model reduces inference computation (e.g., running time and memory use) compared against the original large model. Compared to other model compression methodologies such as pruning (Ye et al., 2018) and quantization (Han et al., 2015), KD has the flexibility of using different architectures of the `student`, which is favored by specific real-world applications.

**Status quo.** Treating `teacher` features as knowledge, KD distills such knowledge to train `student` by encouraging its features to be similar to the `teacher`'s. Through the lens of image classification, prevailing methods can be categorized into two types: logit distillation (Fig 1-left), and feature distillation (Fig 1-right). Logit distillation trains the `student` by minimizing the KL divergence between its logits and the `teacher`'s (Hinton et al., 2015; Zhao et al., 2022). It assumes that, if `student` can produce logits more similar to `teacher`'s, it should achieve better performance, approaching `teacher` performance. However, logit distillation consider only the logit layer but not other intermediate layers. To exploit such, feature distillation trains `student` by encouraging its intermediate-layer features to be similar to the `teacher`'s, e.g., by minimizing the L2 distance between their features (Chen et al., 2021; Zagoruyko & Komodakis, 2016a).

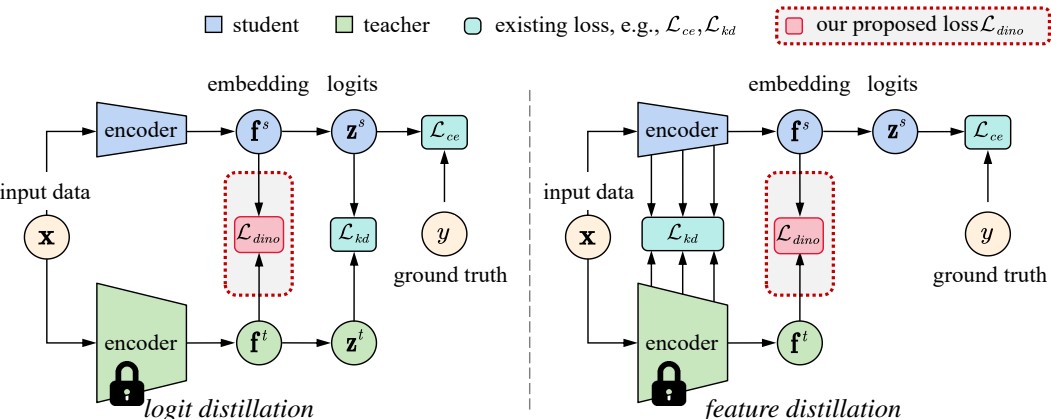

Figure 1: Our main contribution is a simple loss, termed $\mathcal{L}_{dino}$, that regularizes the **di**rection and **no**rm of the `student` features (details in Sec. 3.3). $\mathcal{L}_{dino}$ is applicable to different KD methods which can be categorized into two types in the context of classification: (left) logit distillation that regularizes logits or softmax scores (e.g., KD (Hinton et al., 2015) and DKD (Zhao et al., 2022)), and (right) feature distillation that regularizes features other than logits (e.g., ReviewKD (Chen et al., 2021)). In this work, we apply $\mathcal{L}_{dino}$ to the embedding feature particularly at the penultimate layer (before logits). Experiments show that learning with $\mathcal{L}_{dino}$ improves existing KD methods, achieving the state-of-the-art benchmarking results for image classification (Table 1) and object detection (Table 3).

**Motivation.** Despite the promising results of logit distillation and feature distillation methods, we note that forcing the `student` to produce similar logits or features to the `teacher`'s does not directly serve the final task, e.g., classification. For example, minimizing the L2 distance between the penultimate-layer features (used to compute logits for classification) does not necessarily help learn a better `student`-classifier. Rather, `student` features are better regularized by the `teacher` to facilitate learning a better `student` classifier. Therefore, we present a simple method that uses `teacher` class-mean features to align `student` features to help learn its classifier. Moreover, we empirically find that encouraging the `student` to produce large-norm features yields better performance (Fig. 2). This echoes other lines of work such as domain adaptation (Xu et al., 2019) and pruning (Ye et al., 2018). This motivates us to train the `student` to produce large-norm features.

**Contributions.** We make three main contributions. First, we take a novel perspective to improve KD by regularizing `student` to produce features that (1) are aligned with class-means features computed by the `teacher`, and (2) have sufficiently large *norm*s. Second, we study multiple baseline methods to achieve such regularizations. We show that when incorporating either or both, existing KD methods yields better performance, e.g., classification accuracy and object detection precision by the `student`. Third, we propose a novel and simple loss that simultaneously regularizes feature **di**rection and **no**rm, termed *dino-loss*. Experiments demonstrate that additionally adopting our dino-loss helps existing KD methods achieve better performance. For example, on the standard benchmark ImageNet (Deng et al., 2009), applying dino-loss to KD (Hinton et al., 2015) achieves 72.49% classification accuracy (Fig. 5 and Table B2), better than the original KD (71.35%), with ResNet-18 and ResNet-50 architectures for `student` and `teacher`, respectively. This outperforms recent methods ReviewKD (Chen et al., 2021) (71.09%) and DKD (Zhao et al., 2022) (71.85%).

## 2  RELATED WORK

**Knowledge distillation** (KD) aims to train a small `student` model by distilling knowledge of a well-trained large `teacher` model. The knowledge is delivered by features produced by the `teacher` for training data. Therefore, the key to KD is to align `student` features to the `teacher`'s. The seminal KD method (Hinton et al., 2015) propose to train `student` by aligning its logits with the `teacher`'s, i.e., minimizing the Kullback-Leibler divergence (KL) between logits. Other works improve KD by decoupling the KL loss into separate meaningful parts (Zhao et al., 2022) or consider logits rankings (Huang et al., 2022). Distilling logit knowledge alone may not be sufficient as this does not exploit intermediate-layer features. Hence, feature distillation propose to align more features at other layers (Romero et al., 2014; Zagoruyko & Komodakis, 2016a; Yim et al., 2017; Heo et al., 2019; Passalis & Tefas, 2018a; Park et al., 2019; Tian et al., 2019; Chen et al., 2021; Beyer et al.,

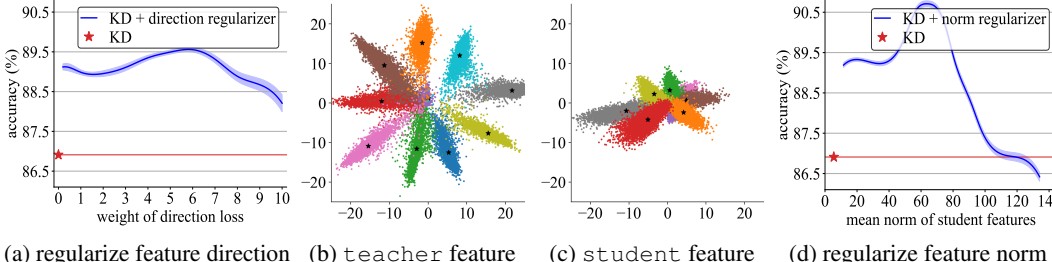

(a) regularize feature direction    (b) `teacher` feature    (c) `student` feature    (d) regularize feature norm

Figure 2: **Regularizing feature direction and norm help knowledge distillation and improves** `student`'**s performance.** We train a Res56 `teacher` and use the KD method (Hinton et al., 2015) to train a Res8 `student` on the CIFAR10 dataset. (a) We propose to regularize `student` by aligning its feature direction with that of class-mean features computed by `teacher`. To do so, we adopt a simple cosine loss term on the penultimate-layer features (Sec. 3.2.1). Results show that regularizing feature direction improves `student`'s performance. (b-c) We train `teacher` and `student` to produce 2D features at the penultimate layer using method KD (Hinton et al., 2015). We visualize them as 2D points, colored by class labels, and mark the class center by ⋆. Notably, the large `teacher` model produces large-norm features (b), while the small `student` model produces small-norm features (c). (d) This motivates us to regularize `student` by encouraging it to produce large-norm features (Sec. 3.2.2). To do so, we use the method SIFN (Xu et al., 2019). Results show that properly regularizing feature norms improves `student`'s performance. Our technical contribution is a simple loss that simultaneously regularize `student` feature **di**rection and **no**rm, so we call this loss **dino**-loss.

2022). In this work, we take a different perspective to improve KD by encouraging the `student` to produce features (at the penultimate layer before logits) that are aligned with the direction of `teacher` classifier and have large norms to improve generalization.

**Constructing classifiers using off-the-shelf features.** Off-the-shelf features extracted from a well-trained model can be used to construct strong classifiers. One simple classifier is to compute class-mean of training examples in the feature space, and uses such as the classifier (Donahue et al., 2014; Sharif Razavian et al., 2014; Kong & Ramanan, 2021). On the other hand, recent literature of pretrained large models (Radford et al., 2021) shows that using off-the-shelf features and cosine similarity is a powerful classifier for zero-shot recognition. In this work, we propose to regularize `student` features using class-mean of `teacher` features. We hypothesize that doing so helps learn better `student` classifiers. Our experiments justify this hypothesis (Table 4).

**Learning large-norm features.** Multiple lines of work find it important to learn large-norm features or weight parameters. For example, domain adaptation (Xu et al., 2019) reveals that the erratic discrimination of the target domain mainly stems from its much smaller feature norms w.r.t that of the source domain, and adopting a larger-norm constraint helps adapt a pretrained model (in the source domain) to a new target domain. Moreover, model pruning finds that features with smaller norms play a less informative role during the inference (Ye et al., 2018), so it is safe to remove weight parameters that produce small-norm features without causing notable performance drop. In our work, we also empirically find that a small-capacity model produces features that tend to collide in the small-norm region (Fig. 2c). Therefore, we are motivated to train `student` to produce large-norm features, hypothesizing that doing so improves `student` performance. Our experiments empirically justify this hypothesis (Fig. 2d, Table 4).

## 3    IMPROVING KNOWLEDGE DISTILLATION BY REGULARIZING FEATURE *DI*RECTION & *NO*RM (DINO)

We describe notations and motivate our study of regularizing feature direction and norm to improve KD. Then, we introduce baselines, followed by our proposed dino-loss.

### 3.1    NOTATIONS AND BACKGROUND

**Notations.** Without losing generality, we think of a classification neural network as two modules: a feature extractor $f(\cdot; \Theta)$, and a classifier $g(\cdot; \mathbf{w})$, which are parameterized by $\Theta$ and $\mathbf{w}$, respectively. For the `teacher`, given input data $\mathbf{x}$, we denote its embedding feature as $\mathbf{f}^t = f^t(\mathbf{x}; \Theta^t)$, and the logits as $\mathbf{z}^t = g^t(\mathbf{f}^t; \mathbf{w}^t)$. Similarly, the `student` outputs the embedding features for $\mathbf{x}$ as

$\mathbf{f}^s = f^s(\mathbf{x}; \Theta^s)$ and logits as $\mathbf{z}^s = g^s(\mathbf{f}^s; \mathbf{w}^s)$. We compute softmax scores in a vector $\mathbf{q}^t = \mathrm{softmax}(\mathbf{z}^t; \tau)$, where $\tau$ is a temperature (default value as 1). Given $N$ training examples from $C$ classes, $\mathbf{x}_i$ and its label $y_i$ (where $i = 1, \dots, N$), we train a classification model (e.g., the `teacher`) by minimizing the cross-entropy (CE) loss $\mathcal{L}_{ce}$ on all the training data.

**Logit distillation** trains the `student` by transferring the `teacher` knowledge using both the CE loss $\mathcal{L}_{ce}$ and a KD loss $\mathcal{L}_{kd}$. The seminal work of KD (Hinton et al., 2015) uses KL divergence as the KD loss $\mathcal{L}_{kd}$, i.e., $\mathcal{L}_{kd} = \frac{1}{N} \sum_{i=1}^{N} \mathbf{KL}(\mathbf{q}_i^t, \mathbf{q}_i^s)$.

**Feature distillation** distills `teacher` knowledge by minimizing the difference of intermediate features at more layers other than the logits (Zagoruyko & Komodakis, 2016a; Yim et al., 2017; Chen et al., 2021). A typical loss term is the L2 distance $\mathcal{L}_2$ between `student` and `teacher` features.[1] For example, over the embedding features at the penultimate layer (before logits), it applies the L2 loss $\mathcal{L}_{kd} = \frac{1}{N} \sum_{i=1}^{N} \mathcal{L}_2(\mathbf{f}_i^s, \mathbf{f}_i^t)$ in addition to the CE loss $\mathcal{L}_{ce}$.

The final loss for KD is $\mathcal{L} = \mathcal{L}_{ce} + \alpha \mathcal{L}_{kd}$, where $\alpha$ controls the significance of the KD loss $\mathcal{L}_{kd}$ depending on distillation choice: either logit distillation or feature distillation.

### 3.2 BASELINE METHODS OF REGULARIZING FEATURE DIRECTION AND NORM

Recall that we are motivated to regularize `student` features during training: aligning their direction with `teacher` class-mean features, and encouraging them to be large in norm. We focus on the embedding features $\mathbf{f}^s$ at the penultimate layer, which are the direct input to a classifier. We compute the class-mean of the $k^{th}$ class as $\mathbf{c}_k = \frac{1}{|\mathcal{I}_k|} \sum_{j \in \mathcal{I}_k} \mathbf{f}_j^t$, where $\mathcal{I}_k$ is the set of indices of training examples belonging to class-$k$. We now introduce simple techniques to regularize `student` features using $\mathbf{c}_k$ in terms of feature direction and norm.

#### 3.2.1 FEATURE DIRECTION REGULARIZATION

We present two simple methods below to regularize `student` w.r.t feature direction.

**Cosine similarity.** We use a simple cosine similarity based loss term to regularize the feature direction of $\mathbf{f}_i^s$ according to its corresponding class-mean $\mathbf{c}_k$:

$$\mathcal{L}_d = \frac{1}{C} \sum_{k=1}^{C} \frac{1}{|\mathcal{I}_k|} \sum_{i \in \mathcal{I}_k} (1 - \cos(\mathbf{f}_i^s, \mathbf{c}_k)) \tag{1}$$

**InfoNCE.** Using the cosine similarity loss Eq. 1 considers only paired examples and their corresponding class-mean. Inspired by InfoNCE (Oord et al., 2018), we also consider inter-class examples and class-means. Therefore, we train `student` by also minimizing:

$$\mathcal{L}_d = \frac{1}{C} \sum_{k=1}^{C} \frac{1}{|\mathcal{I}_k|} \sum_{i \in \mathcal{I}_k} -\log \frac{\exp\left(\cos(\mathbf{f}_i^s, \mathbf{c}_k)\right)}{\sum_{j=1}^{C} \exp\left(\cos(\mathbf{f}_i^s, \mathbf{c}_j)\right)} \tag{2}$$

#### 3.2.2 FEATURE NORM REGULARIZATION

We present two methods below to regularize `student` towards producing large-norm features.

$\mathcal{L}_2$ **distance**. As shown by Fig. 2c, the small-capacity `student` model produces features that have notably smaller norm than the `teacher`'s. To train the `student` to produce larger-norm features, perhaps a naive method is to increase `student` feature norm towards `teacher`'s. To this end, we minimize the L2 distance between features of `student` and `teacher`:

$$\mathcal{L}_n = \frac{1}{C} \sum_{k=1}^{C} \frac{1}{|\mathcal{I}_k|} \sum_{i \in \mathcal{I}_k} \|\mathbf{f}_i^s - \mathbf{f}_i^t\|_2^2 \tag{3}$$

While minimizing Eq. 3 is a common practice in feature distillation, it implicitly trains `student` to produce features with norms approaching the corresponding larger-norm `teacher` features.

---

[1] When features of `student` and `teacher` have different dimensions, one can learn extra modules along with `student` to project its features to the same dimension as `teacher`'s (Chen et al., 2021).

**Stepwise increasing feature norms (SIFN).** We now describe a loss to explicitly increase the norm of the `student` features. Inspired by Xu et al. (2019), we gradually increase the feature norm by minimizing:

$$\mathcal{L}_n = \frac{1}{N} \sum_{i=1}^{N} \mathcal{L}_2 \left( f^s(\mathbf{x}_i; \Theta^s_{previous}) + r, f^s(\mathbf{x}_i; \Theta^s_{current}) \right) \tag{4}$$

where $\Theta^s_{previous}$ and $\Theta^s_{current}$ are parameters of an early checkpoint and the current model being optimized, respectively; $r$ is a step size to increase the norm of `student` features during training.

### 3.3 THE PROPOSED DINO LOSS

For simplicity, we drop the subscript (i.e., the index of a training example or class ID). Let $\mathbf{f}^s$ and $\mathbf{f}^t$ be the embedding features of an input example $\mathbf{x}$ computed by `student` and `teacher`, respectively. Based on $\mathbf{x}$'s ground-truth label $y$, we have its corresponding class-mean $\mathbf{c}$. We compute the projection of $\mathbf{f}^s$ along the direction of $\mathbf{c}$: $\mathbf{p}^s = \mathbf{e} \|\mathbf{f}^s\|_2 \cos(\mathbf{f}^s, \mathbf{c})$. We denote the unit vector $\mathbf{e} = \mathbf{c}/\|\mathbf{c}\|_2$, and $\mathbf{p}^t = \mathbf{e} \|\mathbf{f}^t\|_2$. For physical meaning, please refer to Fig. 3.

When the norm of $\mathbf{f}^s$ is small, or its projection $\mathbf{p}^s$ has small norm, i.e., $\|\mathbf{p}^s\|_2 < \|\mathbf{f}^t\|_2$, we encourage the `student` to output larger-norm features and align them with the `teacher` class-mean by minimizing $\|\mathbf{p}^t - \mathbf{p}^s\|_2$. Because the feature norms of different examples can vary by an order of magnitude (see Fig. 2c), naively learning with the above can produce artificially large gradients from specific training data and negatively affect training. Thus, we divide the above by $\|\mathbf{f}^t\|_2$, which is equivalent to $\|\mathbf{p}^t\|_2$:

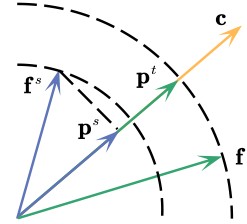

Figure 3: Illustration of notations used in our dino-loss loss.

$$\mathcal{L}_{dino} = \frac{\|\mathbf{p}^t - \mathbf{p}^s\|_2}{\|\mathbf{f}^t\|_2} = \frac{\|\mathbf{p}^t\|_2 - \|\mathbf{p}^s\|_2}{\|\mathbf{f}^t\|_2} = 1 - \frac{\mathbf{f}^s \cdot \mathbf{e}}{\|\mathbf{f}^t\|_2} \tag{5}$$

Minimizing Eq. 5 amounts to simultaneously (1) increasing the norm of $\mathbf{f}^s$ and (2) reducing the angular distance between $\mathbf{f}^s$ and the class-mean $\mathbf{c}$.

When the norm of $\mathbf{f}^s$ is large, i.e., $\|\mathbf{f}^s\|_2 \geq \|\mathbf{f}^t\|_2$, we only minimize the angular distance between `student` feature and the class-mean defined by the `teacher`:

$$\mathcal{L}_{dino} = 1 - \frac{\mathbf{f}^s \cdot \mathbf{e}}{\|\mathbf{f}^s\|_2} \tag{6}$$

The above loss means that the feature norm of `student` is no longer explicitly required to reach a larger value if it is alreayd large enough; yet it is still allowed to increase freely during training. We merge Eq. 5 and 6 and average over all training examples as our dino-loss (dropping constant 1):

$$\mathcal{L}_{dino} = -\frac{1}{C} \sum_{k=1}^{C} \frac{1}{|\mathcal{I}_k|} \sum_{i \in \mathcal{I}_k} \frac{\mathbf{f}_i^s \cdot \mathbf{e}_k}{\max \{\|\mathbf{f}_i^s\|_2, \|\mathbf{f}_i^t\|_2\}} \tag{7}$$

Compatible with existing KD methods, our dino-loss $\mathcal{L}_{dino}$ can be used altogether with CE loss $\mathcal{L}_{ce}$ and KD loss $\mathcal{L}_{kd}$ to train `student`:

$$\mathcal{L} = \mathcal{L}_{ce} + \alpha \mathcal{L}_{kd} + \beta \mathcal{L}_{dino} \tag{8}$$

$\alpha$ and $\beta$ are the weights for $\mathcal{L}_{kd}$ and $\mathcal{L}_{dino}$, respectively. The definition of $\mathcal{L}_{kd}$ depends on the distillation method. Otherwise stated, we study $\mathcal{L}_{dino}$ with the seminal logit distillation method KD (Hinton et al., 2015), so $\mathcal{L}_{kd} = \mathbf{KL}$.

**Remark.** The dino-loss simultaneously encourages `student` to output large-norm features (even larger than `teacher`'s, as shown in Fig. 4), and directly minimizes the angular distance between `student` features and the class-mean defined by the `teacher` during training. This is a desired property in terms of training the `student` to achieve better classification accuracy.

## 4 EXPERIMENTS

In this section, we explore the efficacy of the proposed regularization techniques, specifically focusing on their impact on feature direction and norms. Sec. 4.1 describes datasets and part of the implementation details. Sec. 4.2 benchmarks our approaches and existing KD methods. Sec. 4.3 ablates our dino-loss with in-depth analyses.

Table 1: Benchmarking results on the CIFAR100 dataset. Methods are reported with top-1 accuracy (%) on test set. **++** means that we apply the proposed dino-loss to existing methods. Clearly, doing so improves performance over the original KD methods and outperforms prior KD methods. We mark performance gains using superscripts in blue.

| Methods | Homogeneous architectures | | | Heterogeneous architectures | | |
|---|---|---|---|---|---|---|
| | ResNet-56 | WRN-40-2 | ResNet-32×4 | ResNet-50 | ResNet-32×4 | ResNet-32×4 |
| | ResNet-20 | WRN-40-1 | ResNet-8×4 | MobileNet-V2 | ShuffleNet-V1 | ShuffleNet-V2 |
| teacher (T) | 72.34 | 75.61 | 79.42 | 79.34 | 79.42 | 79.42 |
| student (S) | 69.06 | 71.98 | 72.50 | 64.60 | 70.50 | 71.82 |
| *Feature distillation methods* | | | | | | |
| FitNet (Romero et al., 2014) | 69.21 | 72.24 | 73.50 | 63.16 | 73.59 | 73.54 |
| RKD (Park et al., 2019) | 69.61 | 72.22 | 71.90 | 64.43 | 72.28 | 73.21 |
| PKT (Passalis & Tefas, 2018b) | 70.34 | 73.45 | 73.64 | 66.52 | 74.10 | 74.69 |
| OFD (Heo et al., 2019) | 70.98 | 74.33 | 74.95 | 69.04 | 75.98 | 76.82 |
| CRD (Tian et al., 2019) | 71.16 | 74.14 | 75.51 | 69.11 | 75.11 | 75.65 |
| ReviewKD (Chen et al., 2021) | 71.89 | 75.09 | 75.63 | 69.89 | 77.45 | 77.78 |
| *Logit distillation methods* | | | | | | |
| KD (Hinton et al., 2015) | 70.66 | 73.54 | 73.33 | 67.65 | 74.07 | 74.45 |
| DIST (Huang et al., 2022) | 71.78 | 74.42 | 75.79 | 69.17 | 75.23 | 76.08 |
| DKD (Zhao et al., 2022) | 71.97 | 74.81 | 75.44 | 70.35 | 76.45 | 77.07 |
| **KD++** | $\mathbf{72.53}^{+1.87}$ | $74.59^{+1.05}$ | $75.54^{+2.21}$ | $70.10^{+2.35}$ | $75.45^{+1.38}$ | $76.42^{+1.97}$ |
| **DIST++** | $72.52^{+0.74}$ | $75.00^{+0.58}$ | $76.13^{+0.34}$ | $69.80^{+0.63}$ | $75.60^{+0.37}$ | $76.64^{+0.56}$ |
| **DKD++** | $72.16^{+0.19}$ | $75.02^{+0.21}$ | $\mathbf{76.28}^{+0.84}$ | $\mathbf{70.82}^{+0.47}$ | $77.11^{+0.66}$ | $77.49^{+0.42}$ |
| **ReviewKD++** | $72.05^{+0.16}$ | $\mathbf{75.66}^{+0.57}$ | $76.07^{+0.44}$ | $70.45^{+0.56}$ | $\mathbf{77.68}^{+0.23}$ | $\mathbf{77.93}^{+0.15}$ |

## 4.1 SETTINGS

We conduct experiments to validate the proposed regularization techniques on feature direction and norms in the context of image classification and object detection, including:

**CIFAR-100** (Krizhevsky et al., 2009) contains 50k training images and 10k testing images. For each input image, 4 pixels are added as padding on each side, and a $32 \times 32$ cropping patch is randomly selected from the padded images or their horizontally flipped counterparts.

**ImageNet** (Russakovsky et al., 2015) comprises 1.28 million training images and 50,000 validation images spanning by 1,000 categories.

**MS-COCO** (Lin et al., 2014) consists of 80 object categories with 118k training images and 5k validation images.

Our implementation adheres to the established conventions within the field, as in prior works such as (Tian et al., 2019; Chen et al., 2021; Zhao et al., 2022; Huang et al., 2022). More details are attached in Appendix A due to the page limit.

## 4.2 COMPARISONS WITH STATE-OF-THE-ART RESULTS

**CIFAR-100 Classification**. Table 1 showcases the performances of knowledge distillation on the CIFAR-100 dataset. In this context, spanning homogeneous and heterogeneous architectures, we undertake an extensive assessment over prominent *feature distillation methods* (e.g., FitNet (Romero et al., 2014), RKD (Park et al., 2019), PKT (Passalis & Tefas, 2018b), OFD (Heo et al., 2019), CRD (Tian et al., 2019), ReviewKD (Chen et al., 2021)) and *logits distillation methods*(e.g., KD (Hinton et al., 2015), DIST (Huang et al., 2022), DKD (Zhao et al., 2022)). The **++** signifies the integration of our novel dino-loss into the preexisting methodologies. A salient conclusion from Table 1 is that *our proposed dino-loss manifests exceptional flexibility, which delivers advancements for both feature and logits distillation methods, irrespective of the homogeneity or heterogeneity for network architectures*. This phenomenon underscores the robust generalization prowess exhibited by the dino-loss within the realm of knowledge distillation.

**ImageNet Classification**. We delve deeper into the efficacy of the proposed dino-loss on the more expansive ImageNet dataset. Table 2 provides supplementary evidence of the flexibility. Remarkably,

Table 2: Benchmarking results on the ImageNet dataset. Methods are reported with top-1 accuracy (%). "T $\rightarrow$ S" marks the architectures of `teacher` and `student`, short for knowledge distillation from the former to the latter. R{18,34,50} are the ResNet18, ResNet34, and ResNet50, respectively. MV1 means MobileNet-V1. Again, additionally using our dino-loss, methods such as KD, ReviewKD, and DKD obtain better performance than their counterparts, achieving the state-of-the-art performance on this dataset.

| T$\rightarrow$S | teacher | student | CRD Tian et al. | SRRL Yang et al. | ReviewKD Chen et al. | KD Hinton et al. | DKD Zhao et al. | KD++ | ReviewKD++ | DKD++ |
|---|---|---|---|---|---|---|---|---|---|---|
| R34$\rightarrow$R18 | 73.31 | 69.76 | 71.17 | 71.73 | 71.62 | 70.66 | 71.70 | 71.98 | 71.64 | **72.07** |
| R50$\rightarrow$MV1 | 76.16 | 68.87 | 71.37 | 72.49 | 72.56 | 70.50 | 72.05 | 72.77 | **72.96** | 72.63 |

Table 3: Detection results (mAP in %) on the **COCO `val2017`** using Faster R-CNN detector. Incorporating our dino-loss, KD++ and ReviewKD++ obtain performance gains over their original counterparts, achieving the state-of-the-art KD performance.

| Method | R101$\rightarrow$R18 | | | R101$\rightarrow$R50 | | | R50$\rightarrow$MV2 | | |
|---|---|---|---|---|---|---|---|---|---|
| | mAP | $AP^{50}$ | $AP^{75}$ | mAP | $AP^{50}$ | $AP^{75}$ | mAP | $AP^{50}$ | $AP^{75}$ |
| `teacher` | 42.04 | 62.48 | 45.88 | 42.04 | 62.48 | 45.88 | 40.22 | 61.02 | 43.81 |
| `student` | 33.26 | 53.61 | 35.26 | 37.93 | 58.84 | 41.05 | 29.47 | 48.87 | 30.90 |
| KD (Hinton et al., 2015) | 33.97 | 54.66 | 36.62 | 38.35 | 59.41 | 41.71 | 30.13 | 50.28 | 31.35 |
| FitNet (Romero et al., 2014) | 34.13 | 54.16 | 36.71 | 38.76 | 59.62 | 41.80 | 30.20 | 49.80 | 31.69 |
| FGFI (Wang et al., 2019) | 35.44 | 55.51 | 38.17 | 39.44 | 60.27 | 43.04 | 31.16 | 50.68 | 32.92 |
| DKD (Zhao et al., 2022) | 35.05 | 56.60 | 37.54 | 39.25 | 60.90 | 42.73 | 32.34 | 53.77 | 34.01 |
| ReviewKD (Chen et al., 2021) | 36.75 | 56.72 | 34.00 | 40.36 | 60.97 | 44.08 | 33.71 | 53.15 | 36.13 |
| **KD++** | 36.12 | 56.81 | 37.64 | 39.86 | 61.07 | 43.57 | 33.26 | 53.71 | 34.85 |
| **ReviewKD++** | **37.43** | **57.96** | **40.15** | **41.03** | **61.80** | **44.94** | **34.51** | **55.18** | **37.21** |

despite its inherent simplicity, our **KD++** approach, which seamlessly integrates the dino-loss into the naive **KD** framework, competes head-to-head with the SOTA results (**KD++** *vs.* (**ReviewKD**, **DKD**). Even, it surpasses the existing leading benchmarks on the extensive ImageNet dataset (Table 2), achieving notable improvements (**KD++**$_{R34\rightarrow R18}$: 71.98% *vs.* **SRRL**$_{R34\rightarrow R18}$: 71.73%, **KD++**$_{R50\rightarrow MV1}$: 72.77% *vs.* **ReviewKD**$_{R50\rightarrow MV1}$: 72.56%).

**COCO Object Detection**. We verify the efficacy of the proposed dino-loss in knowledge distillation for object detection tasks on the COCO dataset, as shown in Table 3. Specifically, the **ReviewKD++** yields a significant improvement in performance, outperforming state-of-the-art results with a remarkable margin.

## 4.3 ABLATION STUDY

In this subsection, we first investigate the ablation experiments on CIFAR-100 pertaining to feature norm and direction regularization. Subsequently, we perform a visual analysis of the impact before and after applying dino-loss. Finally, we conduct intriguing experiments on ImageNet and observe that our approach accrues advantages from employing larger `teacher` models.

**The isolation of feature direction and norm regularization.** Recall that Sec. 3.2.1 and Sec. 3.2.2 explore the concrete instantiation of feature direction and norm regularization separately. Owing to space limitations, we present only simple test results for $\mathcal{L}_2$ (Eq. 3) and SIFN (Eq. 4) on CIFAR-100 in Table 4b. Yet additional offline experiments substantiate that SIFN outperforms $\mathcal{L}_2$ regularization in terms of performance and consistently affirm that large `student` feature norms encapsulate more `teacher` knowledge. Similarly, Table 4a demonstrates the superior gains of cosine (Eq. 1) compared to InfoNCE (Eq. 2), further underscoring the significance of feature direction constraints.

**DINO loss yields better results.** We discuss the benefits of the independent amalgamation of feature norm and direction regularization. Table 4c consolidates feature direction regularization (cosine, InfoNCE) and feature norm ($\mathcal{L}_2$, SIFN), unveiling that the optimal setting (cosine + SIFN) leads to superior performance (69.07%) among all combinations. Nevertheless, upon meticulous scrutiny, it becomes apparent that directly integrating feature direction with norm regularization can prove deleterious, as it engenders lower results than separate regularization. For instance, (cosine + $\mathcal{L}_2$) or (cosine + SIFN) reduces accuracy from 69.18% to 68.62% (-0.56%) and 69.07% (-0.11%), respectively. Similarly, (SIFN + cosine) or (SIFN + InfoNCE) results in a substantial decline from

Table 4: **Analysis of feature direction and norm regularization.** We train `teacher` (ResNet-50) and `student` (MobileNet-V2) models on the CIFAR100 dataset and report accuracy (%) on its test-set. We use KD (Hinton et al., 2015) as the *baseline*, which is a logit distillation method. From (a-b), we see that applying either direction or norm regularization on `student` features improves KD as shown by the increased `student` accuracy. While combining both outperforms *baseline* (c), using dino-loss achieves the best (d).

(a) Regularizing feature direction only.

| case | R50-MV2 | R56-R20 |
|---|---|---|
| *baseline* | 67.65 | 70.66 |
| cosine | 69.18 | 71.75 |
| InfoNCE | 69.06 | 70.73 |

(b) Regularizing feature norm only.

| case | acc. |
|---|---|
| *baseline* | 67.65 |
| $\mathcal{L}_2$ | 69.05 |
| SIFN | 69.32 |

(c) Regularizing both feature norm and direction.

| case | acc. |
|---|---|
| cosine + $\mathcal{L}_2$ | 68.62 |
| cosine + SIFN | 69.07 |
| InfoNCE + $\mathcal{L}_2$ | 68.47 |
| InfoNCE + SIFN | 68.71 |

(d) The proposed dino-loss works the best.

| case | acc. |
|---|---|
| CE + KL (*baseline*) | 67.65 |
| CE + DINO | 68.78 |
| KL + DINO | 68.68 |
| CE + KL + DINO | **70.10** |

Table 5: Comparison of using `teacher`'s classifier weights (dubbed "w/ weights") versus per-class mean features (dubbed "w/ class-mean") in our dino-loss. We study them with the KD method (Hinton et al., 2015) on the CIFAR-100 dataset. Results show that using per-class mean features outperforms classifier weights.

| Methods | Homogeneous architectures | | | Heterogeneous architectures | | |
|---|---|---|---|---|---|---|
| | ResNet-56 | WRN-40-2 | ResNet-32×4 | ResNet-50 | ResNet-32×4 | ResNet-32×4 |
| | ResNet-20 | WRN-40-1 | ResNet-8×4 | MobileNet-V2 | ShuffleNet-V1 | ShuffleNet-V2 |
| `teacher` | 72.34 | 75.61 | 79.42 | 79.34 | 79.42 | 79.42 |
| `student` | 69.06 | 71.98 | 72.50 | 64.60 | 70.50 | 71.82 |
| KD (Hinton et al., 2015) | 70.66 | 73.54 | 73.33 | 67.65 | 74.07 | 74.45 |
| L2 of cls weights | 70.54 | 73.61 | 73.76 | 66.81 | 73.62 | 74.13 |
| w/ weights | 71.73 | 73.97 | 75.06 | 69.76 | 75.24 | 75.61 |
| **w/ class-mean** | **72.53** | **74.59** | **75.54** | **70.10** | **75.45** | **76.42** |

69.32% to 69.07% (-0.25%) and 68.71% (-0.61%), respectively. In contrast, the proposed dino-loss exploits both strategies, yielding the best performance at 70.10% accuracy (Table 4d).

**Class-mean *vs.* classifier weights.** We perform a quantitative analysis of the classifier weights and per-class feature centers. We adopt the classifier weights that are derived from `teacher` as centers in our dino-loss to train `student` models (dubbed "w/ weights"). In comparison, we utilize per-class feature centers, denoted as "w/ class-mean" (which is our proposed method). The results presented in Table 5 clearly demonstrate that the utilization of per-class feature centers surpasses the performance achieved by using `teacher`'s classifier weights. Additionally, we have implemented an alternative approach that employs an L2 loss to guide the `student` to output classifier weights similar to those of the `teacher` (referred to as "L2 of cls weight"). However, this approach consistently underperforms our "w/ class-mean" method and even lags behind the baseline KD method (Hinton et al., 2015) in most experimental settings. These findings provide strong evidence for the superiority of employing class-mean features over the `teacher`'s classifier weights.

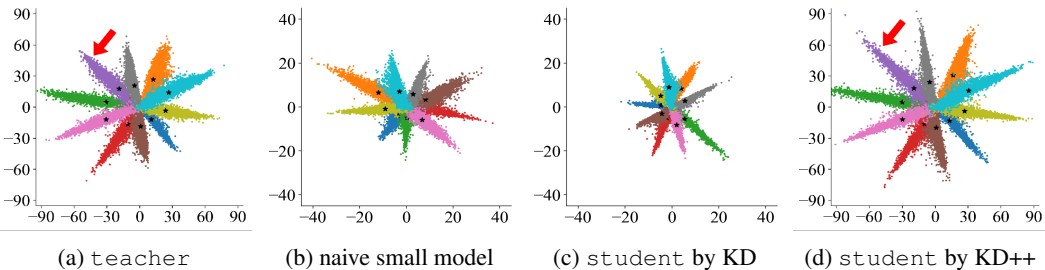

(a) `teacher`    (b) naive small model    (c) `student` by KD    (d) `student` by KD++

Figure 4: **Visualization of 2D embedding features.** (a) Features computed by `teacher` (ResNet-50) are well separated at class label; note the purple class pointed by red arrow. (b) a small-capacity model (ResNet-18) fails to separate this class, which is occluded by others. (c) Even using KD (Hinton et al., 2015) to train ResNet-18 `student` cannot reveal this purple class. (d) Using our dino-loss along with KD, i.e., KD++, achieves better separation of the points and reveals purple class. This attributes to the feature direction regularization using `teacher` class-means. Moreover, `student` features in (d) have larger-norms than the `teacher` in (a).

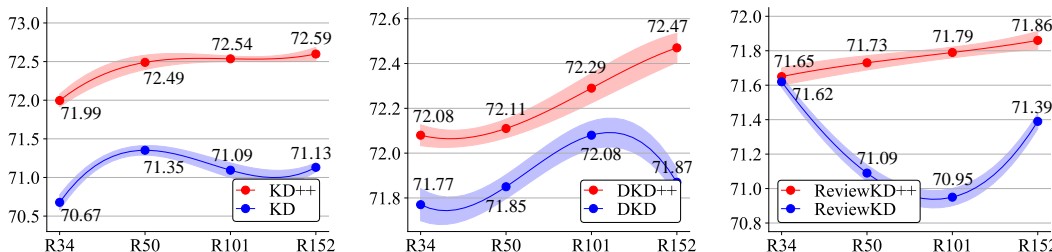

Figure 5: **DINO can benefit from larger teachers.** With the teacher capacity increasing, our method, KD++, DKD++ and ReviewKD++ (red) is able to learn better distillation results, even though the original distillation methods (blue) suffers from degradation problems. The student is ResNet-18, with scaling up the teacher from ResNet-34 to ResNet-152, and reported the Top-1 accuracy (%) on the ImageNet validation set. All results are the average over 5 trials.

**KD++ as a stronger baseline.** Table 4d illustrates the impacts of different losses in canonical knowledge distillation. By incorporating dino-loss into conventional KD framework (Hinton et al., 2015), **KD++** (i.e., CE+KL+DINO) achieves a stunning result (**KD** (67.65%)→ **KD++** (70.10%)). Interestingly, combining dino-loss alone with CE or KL can also boost the accuracy by about 1% compared to classical KD. It is worth noting that **KD++** introduces virtually no additional parameters and minimal computational overhead, making it a stronger baseline for knowledge distillation (more validations can be gleaned from the results presented in Table 1&2&3).

In addition, we visually examine the feature with a learnable dimension reduction approach (Wen et al., 2016), as shown in Fig. 4. First, as indicated in Fig. 4d, **KD++** demonstrates notably amplified feature norms, surpassing even those of the teacher depicted in Fig. 4a. Furthermore, the direction in **KD++** align well with the teacher (Fig. 4a *vs.* Fig. 4d, thereby maintaining consistent relative margins among categories. Another observation is that both the naive student (Fig. 4b) and the conventional KD (Fig. 4c) exhibit direct failures in classifying the purple category, whereas our approach, **KD++** (Fig. 4d), effectively reattends to the "disappeared" category.

**Benefit from larger teacher models.** Since previous experiments highlight that consistent direction with a larger norm for student can better facilitate the assimilation of knowledge from teacher, we further investigate whether our approach exhibits monotonic incremental gains when faced with larger teacher. As shown in Fig. 5, it is evident that for KD (Hinton et al., 2015), DKD (Zhao et al., 2022) and ReviewKD (Chen et al., 2021) show a degradation or fluctuation trend when scaling up the teacher from ResNet-34 to ResNet-152. Surprisingly, upon incorporating our dino-loss, the results showcase a consistent improvement (e.g., **KD++**: 71.99% → 72.49% → 72.54% → 72.59%). In addition, we also experimented with distilling from Transformer (Dosovitskiy et al., 2020) to ResNet in Appendix B.5, and studied the effect of increasing the size of the student on knowledge distillation in Table B3. These results show that KD++ consistently outperforms its competitors by a significant margin across fifferent settings, including: larger teacher, larger student, and heterogeneous architectures. This is a desired property that *simple distillation methods outperform sophisticated ones*.

# 5 DISCUSSION AND CONCLUSION

**Broader Impacts and Limitations.** As our work falls in the area of knowledge distillation, we do not see any new potential societal impacts other than those already known, e.g., student models might learn bias and unfairness delievered by the teacher. Our work has some visible limitations, e.g., we apply dino-loss to the penultimate layer only, and we do not study how to distill large pretrained models (e.g., language models). Addressing these are important and future work.

**Conclusion.** We study feature regularization w.r.t norm and direction when training student models for better knowledge distillation (KD). Indeed, experiments demonstrate that doing so with our explored simple methods and the proposed dino-loss helps existing KD methods achieve better performance. We expect the proposed dino-loss to be a plug-in in future KD methods.

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

# Appendix

## A MORE IMPLEMENTATION DETAILS

For fair comparisons, our implementation adheres to the previous methodologies outlined in (Tian et al., 2019; Chen et al., 2021; Zhao et al., 2022; Huang et al., 2022). The hyperparameters $\alpha$ and $\beta$ are determined through an exhaustive search conducted within a predefined range, aligning with the established practices in prior studies.

**CIFAR-100** (Krizhevsky et al., 2009) contains 50k training images and 10k testing images. For each input image, 4 pixels are added as padding on each side, and a $32 \times 32$ cropping patch is randomly selected from the padded images or their horizontally flipped counterparts. We employ weight initialization as described in He et al. (2015), training all `student` networks from scratch, while the teachers load the publicly available weights from Tian et al. (2019). The `student` networks are trained using a mini-batch size of 128 over 240 epochs (with a linear warmup for the first 20 epochs), employing SGD with a weight decay of 5e-4 and momentum of 0.9. We set the initial learning rate of 0.1 for ResNet (He et al., 2016) and WRN (Zagoruyko & Komodakis, 2016b) backbones, and 0.02 for MobileNet (Sandler et al., 2018) and ShuffleNet (Ma et al., 2018) backbones, decaying it with a factor of 10 at 150th, 180th, and 210th. The temperature is empirically set to 4.

**ImageNet** (Russakovsky et al., 2015) comprises 1.28 million training images and 50,000 validation images spanning by 1,000 categories. We employ SGD with a mini-batch size of 512 for a total of 100 epochs (with a linear warmup for the first 5 epochs). The initial learning rate is set to 0.2 and is reduced by a factor of 10 every 30 epochs. Besides, the weight decay and momentum are set to 1e-4 and 0.9, respectively. The pre-trained weights for teachers come from PyTorch[2] and TIMM (Wightman, 2019) for fair comparisons. The temperature for knowledge distillation is set to 1.

**COCO** 2017 (Lin et al., 2014) consists of 80 object categories with 118k training images and 5k validation images. We utilize Faster R-CNN (Ren et al., 2015) with FPN (Lin et al., 2017) as the feature extractor, and employ the dino-loss on the R-CNN head, wherein both `teacher` and `student` models adopt ResNet (He et al., 2016). In addition, MobileNet-V2 (Sandler et al., 2018) is used as a heterogeneous `student` model. All `student` models are trained with 1x scheduler, following Detectron2 [3].

Our proposed dino-loss function regularizes the norm and direction of the `student` features at the penultimate layer before logits. The embedding features of the `student` and `teacher` models may have different dimensions. This can be addressed by learning a fully connected layer (followed by Batch Normalization) with the `student` to project its features to the same dimension as the `teacher`'s.

## B ADDITIONAL ABLATION STUDIES

### B.1 THE IMPACT OF HYPER-PARAMETERS $\alpha$ AND $\beta$

In the Eq. 8, we introduce the KD++ loss function as $\mathcal{L} = \mathcal{L}_{ce} + \alpha\mathcal{L}_{kd} + \beta\mathcal{L}_{dino}$. As elucidated in the experiment details, the values of $\alpha$ and $\beta$ are acquired through an exhaustive search within a predefined range. To substantiate the efficacy of the proposed dino-loss, we conduct extensive experiments aiming at probing the sensitivities of the hyperparameters $\alpha$ and $\beta$, as depicted in Fig. A1. The dashed lines illustrate the standard KD loss (corresponding to specific setting($\alpha = 1$ and $\beta = 0$)) in Fig. A1a. Evidently, our proposed dino-loss consistently surpasses the scenario devoid of dino-loss as $\beta$ ranges from 0.5 to 4.0 (the solid line always surpasses the dashed line for the same color). Furthermore, in Fig. A1b, when the optimal $\beta$ value is fixed, the distilled performance exhibits consistent enhancement compared to the baseline as $\alpha$ varies. These results compellingly attest to the overarching efficacy of the proposed dino-loss in our experiments, with the sensitivity of hyperparameters merely influencing the magnitude of improvement.

---

[2]https://pytorch.org/vision/stable/models.html
[3]https://github.com/facebookresearch/detectron2

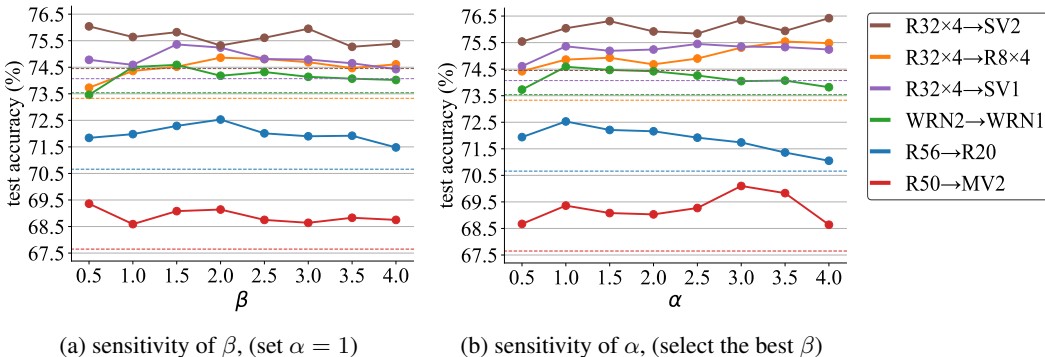

(a) sensitivity of $\beta$, (set $\alpha = 1$)  (b) sensitivity of $\alpha$, (select the best $\beta$)

Figure A1: The impact of hyper-parameters $\alpha$ and $\beta$. The dashed lines illustrate the performance based on standard KD loss (corresponding to the specific setting ($\alpha = 1$ and $\beta = 0$)). (a). $\alpha$ is set to 1, then evaluate the impact of $\beta$. (b). keep best $\beta$ fixed, assessing the impact of $\alpha$.

Certainly, an alternative approach worth contemplating for acquiring optimal parameters entails performing a grid search within the hyperplane spanning by $\alpha$ and $\beta$. Nevertheless, such an approach incurs heightened intricacy and computational demands. The goal of this study, however, resides in substantiating the efficacy of the proposed dino-loss, thereby necessitating the demonstration that outcomes attained with non-zero $\beta$ surpass those achieved through the conventional KD setting ($\alpha$=1 and $\beta$=0). In practical scenarios pertaining to knowledge distillation tasks, it becomes feasible to ascertain the optimal $\alpha$ and $\beta$ parameter pairs by undertaking a grid search across the $\alpha - \beta$ parameter space, while judiciously considering the facet of actual performance augmentation.

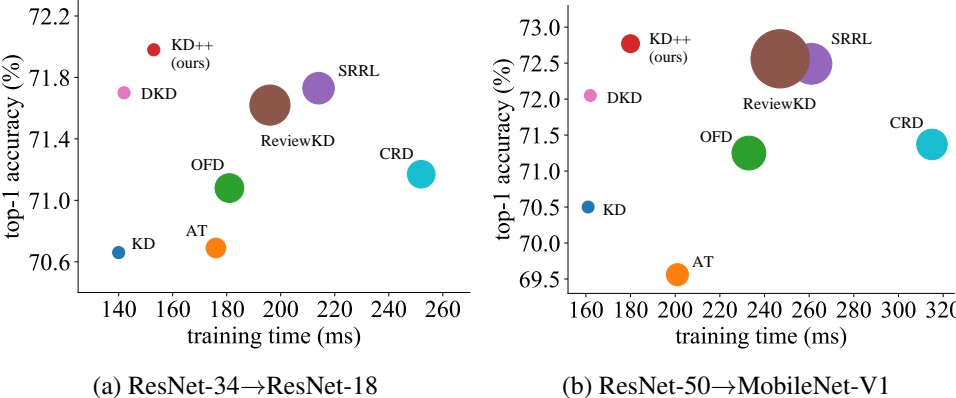

(a) ResNet-34→ResNet-18  (b) ResNet-50→MobileNet-V1

Figure A2: **Wall-clock time per training iteration vs. accuracy** on the ImageNet validation set. left: homogeneous architectures, right: heterogeneous architectures. Enlarged circles correspond to a higher demand for parameters.

## B.2 COMPLEXITY COMPARISONS

In this subsection, we present simple comparisons for mainstream knowledge distillation methods, as illustrated in Fig. A2. Fig. A2a and Fig. A2b showcase examples of homogeneous distillation (ResNet-34 → ResNet-18) and heterogeneous distillation (ResNet-50 → MobileNet-V1) on the ImageNet dataset. We measure the average time cost per batch iteration over the entire dataset as the horizontal axis and the Top-1 accuracy as the vertical axis. The varying sizes of circular markers representing different methods are proportional to the actual model parameter sizes. It is clear that our approach (KD++) delivers better performance with a small amount of time expense. It is important to highlight that in heterogeneous knowledge distillation tasks, there is typically a disparity in feature dimensions. Consequently, the inclusion of a bridging linear dimension transformation layer becomes imperative, attributing to the marginal increment in parameterization observed in our method, KD++, as compared to the classical KD approach.

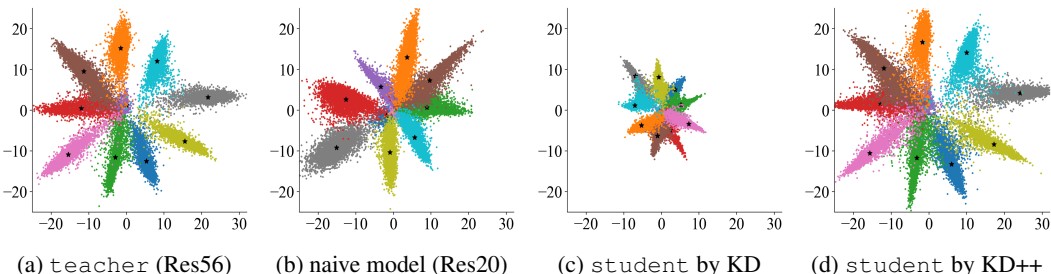

|  (a) teacher (Res56) | (b) naive model (Res20) | (c) student by KD | (d) student by KD++ |

Figure A3: **Embedding features visualization on CIFAR-10.** Teacher and student are ResNet-56 and ResNet-20, respectively. The same color belongs to the same category. ⋆ mean that class centers.

### B.3 MORE VISUALIZATION OF EMBEDDING FEATURES

Although PCA (Pearson, 1901) or t-SNE (Maaten & Hinton, 2008) have proven to be effective nonlinear dimensionality reduction techniques, we still adhere to the common practice of providing a more intuitive understanding. Therefore, we follow the approach of (Wen et al., 2016; Xu et al., 2019) and introduce a 2-dimensional learnable feature output at the feature layer for visual analysis. We select the feature statistics of 10 classes from the teacher and student models on CIFAR-10 and visualize their 2D features, as shown in Fig. A3. Our approach, KD++, clearly demonstrates more intuitive results.

### B.4 DOES THE MAGNITUDE OF TEACHER NORM MATTER ?

In earlier sections, we discover that improving the student's norm benefits knowledge distillation. Therefore, a natural question arises: does increasing the teacher norm also contribute to improving student performance? To investigate this, we conduct simple experiments where we introduce a scaling factor, denoted as $m$, to the norm of the teacher in Eq. 7 as follows:

$$\mathcal{L}_{dino} = -\frac{1}{C} \sum_{k=1}^{C} \frac{1}{|\mathcal{I}_k|} \sum_{i \in \mathcal{I}_k} \frac{\mathbf{f}_i^s \cdot \mathbf{e}_k}{\max\left\{\|\mathbf{f}_i^s\|_2, \|\mathbf{f}_i^t\|_2 \cdot (1+m)\right\}} \tag{9}$$

Interestingly, our experimental results (Table B1) indicate that in the context of homogeneous knowledge distillation, altering the norm of the teacher, whether increasing or decreasing it, does not lead to better improvement in student performance compared to maintaining the original norm of the teacher. However, in the case of heterogeneous knowledge distillation, there may be benefits in appropriately increasing the norm of the teacher features. It is worth noting that since this experiment has not been tested on a large-scale dataset, we cannot definitively conclude whether a larger teacher norm will always result in improvements. Nonetheless, this presents a promising direction for future exploration, where joint constraints on the norm size and direction can be applied to both teacher and student.

Table B1: Altering the norm of the teacher mode with a scaling factor $m$. Classification accuracy on the CIFAR-100 test set. The gray background indicates the default setting.

| $m$ | -0.5 | -0.1 | 0.0 | 0.1 | 0.5 | 0.7 | 1.0 | 1.5 | 2.0 |
|---|---|---|---|---|---|---|---|---|---|
| R56→ R20 | 71.57 | 72.19 | **72.53** | 71.76 | 71.86 | 71.64 | 71.79 | 71.74 | 71.92 |
| R50→ MV2 | 69.46 | 69.43 | 70.10 | 70.17 | **70.23** | 69.68 | 69.72 | 68.49 | 69.44 |

### B.5 EXPERIMENTS WITH LARGER TEACHER AND LARGER STUDENT

Fig. 5 clearly shows that our method can benefit from larger teachers. We report the mean top-1 accuracy on the validation with standard deviation over five runs, and the results of distillation from ViT (Dosovitskiy et al., 2020) to ResNet in Table B2. KD++ consistently outperforms the competitions. Nonetheless, owing to the architectural differences, specifically the contrasting characteristics

Table B2: **Our method could benefit from larger teachers.** Methods are reported with top-1 accuracy (%) on the ImageNet validation set. With `teacher` capacity increasing, `student` models (trained with our dino-loss) achieve better classification results. Yet, previous KD methods do not necessarily obtain better results by distilling larger teachers. ∗ represents our implementation based on the official code. All results are the average over 5 trials. We mark standard deviation using superscripts in blue.

| student | teacher | student | teacher | KD* Hinton et al. | ReviewKD* Chen et al. | DKD* Zhao et al. | KD++ | ReviewKD++ | DKD++ |
|---|---|---|---|---|---|---|---|---|---|
| ResNet-18 | ResNet-34 | 69.76 | 73.31 | $70.68^{\pm0.098}$ | $71.62^{\pm0.031}$ | $71.77^{\pm0.072}$ | $71.99^{\pm0.082}$ | $71.65^{\pm0.051}$ | $\mathbf{72.08}^{\pm0.047}$ |
| | ResNet-50 | | 76.16 | $71.35^{\pm0.062}$ | $71.09^{\pm0.047}$ | $71.85^{\pm0.054}$ | $\mathbf{72.49}^{\pm0.093}$ | $71.73^{\pm0.041}$ | $72.11^{\pm0.042}$ |
| | ResNet-101 | | 77.37 | $71.09^{\pm0.095}$ | $70.95^{\pm0.050}$ | $72.08^{\pm0.063}$ | $\mathbf{72.54}^{\pm0.036}$ | $71.79^{\pm0.031}$ | $72.29^{\pm0.066}$ |
| | ResNet-152 | | 78.31 | $71.13^{\pm0.057}$ | $71.39^{\pm0.044}$ | $71.87^{\pm0.060}$ | $\mathbf{72.59}^{\pm0.086}$ | $71.86^{\pm0.051}$ | $72.47^{\pm0.065}$ |
| ResNet-18 | ViT-S | 69.76 | 74.64 | $71.32^{\pm0.061}$ | n/a | $71.21^{\pm0.068}$ | $\mathbf{71.46}^{\pm0.032}$ | n/a | $71.33^{\pm0.043}$ |
| | ViT-B | | 78.00 | $71.63^{\pm0.054}$ | n/a | $71.62^{\pm0.071}$ | $\mathbf{71.84}^{\pm0.066}$ | n/a | $71.69^{\pm0.075}$ |

of global attention in Transformer and local receptive fields in Convolution, the benefits are not as conspicuous as in cases with homogeneous architectures.

We used KD++ to study the effect of increasing the size of the `student` on knowledge distillation, and set the `teacher` as ResNet-152. The results are shown in the Table B3, and demonstrate that increasing the capacity of the `student` can significantly improve the distillation results, even surpass the `teacher`, such as ResNet-152 distilled to ResNet-101: 78.31% → 79.15%.

Table B3: **Larger students get better distillation.** The teacher is ResNet-152 (top-1 acc, 78.31%), and reported with top-1 accuracy (%) on the ImageNet validation set.

| student | ResNet-18 | ResNet-34 | ResNet-50 | ResNet-101 |
|---|---|---|---|---|
| naive (He et al., 2016) | 69.76 | 73.31 | 76.16 | 77.37 |
| KD (Hinton et al., 2015) | 70.66 | 74.84 | 76.93 | 78.04 |
| **KD++** | **71.98** | **75.53** | **77.48** | **79.15** |

## B.6 THE SAMPLE SELECTION STRATEGY FOR CLASS MEAN

For small-scale datasets such as CIFAR, we compute the mean of the embedded features of samples in the entire training set as the class centers. In practice, these models often suffer from overfitting, achieving close to 100% accuracy on the training set. Therefore, using all samples does not affect the class centers. However, for large-scale datasets like ImageNet, the models exhibit lower accuracy on the training set (e.g., 73.31% for ResNet-34). In such cases, using all training samples to evaluate class centers would inevitably impact the distribution of each class center. We investigate two methods for computing class centers on ImageNet: (1) utilizing all samples and (2) only considering the correctly predicted samples by the `teacher` model. It is important to note that all samples are derived from the training set. The `teacher` and `student` models are ResNet-34 and ResNet-18. We found that the result (72.01%) by only **the correctly predicted samples** by the `teacher` **slightly outperforms using all samples** (71.98%). This confirms the existence of this issue in large-scale datasets; however, the impact is insignificant. Therefore, we default to using all samples for computing class centers.

