# OpenReview forum: "Improving Knowledge Distillation via Regularizing Feature Direction and Norm"
_ICLR.cc/2024/Conference — ICLR 2024 Conference Withdrawn Submission_

### Official Review · Reviewer_HgHQ · 2023-10-13

**Soundness:** 2 fair
**Presentation:** 3 good
**Contribution:** 2 fair
**Rating:** 3
**Confidence:** 5

**Summary:**

The paper proposes a simple yet efficient feature direction distillation loss. Experiments show that this significantly improves KD
performance.

**Strengths:**

1. Improving KD by feature norm and direction is reasonable and effectiveness.
2. Experiments on standard benchmarks demonstrate that adopting $\mathcal{L}_{dino}$ remarkably improves existing KD methods.

**Weaknesses:**

1. The contributions seem a little limited.
2. There is lack of theoretical analysis of DINO loss. The paper is not good enough to be published on ICLR.

**Questions:**

1. How to align the features between heterogeneous architectures?
2. Could you please provide more theoretical analysis?
3. What about extending it to a multi-layer version of feature distillation?
4. How to apply the proposed method to existing KD methods, e.g. ReviewKD, DKD, DIST? Just add the DINO loss function to the total loss ? If so, I think adding other loss like contrastive distillation loss or RKD may also make a improvement.

---

### Official Review · Reviewer_yLjx · 2023-10-13

**Soundness:** 3 good
**Presentation:** 3 good
**Contribution:** 2 fair
**Rating:** 6
**Confidence:** 5

**Summary:**

Here is a summary of the key points from the paper:

- The paper proposes a method to improve knowledge distillation (KD) by regularizing student features to align direction with teacher class-means and have sufficiently large norms.

- Current KD methods like logit or feature distillation align student and teacher but don't directly optimize for student's task performance.

- The paper shows regularizing direction using cosine similarity to teacher class means helps improve student accuracy.

- It also finds student models tend to produce smaller-norm features, so encouraging larger norms improves performance.

- A simple combined loss called dino-loss is proposed to simultaneously regularize student feature direction and norm using teacher class means.

- Experiments on CIFAR and ImageNet classification, and COCO detection show dino-loss consistently improves various KD methods like KD, ReviewKD, DKD.

- Dino-loss achieves new state-of-the-art results among KD techniques on classification and detection benchmarks.

- The method is model-agnostic, simple to implement, adds minimal overhead, and benefits from larger teacher models.

In summary, the key contributions are a way to improve KD by regularizing student features for better alignment and norms, along with a simple and effective dino-loss to achieve this jointly. The results demonstrate consistent gains across tasks and benchmarks.

**Strengths:**

The paper presents an original and significant approach to improve KD via thoughtful feature regularization. The method is intuitive and supported by quality experiments. The gains are demonstrated to be significant across tasks. The presentation and discussion are clear:
- The method and dino-loss are clearly explained with illustrations and equations. Results are well-presented in tables and figures. Limitations are properly discussed.
- Improving KD is an important practical problem. The consistent gains are significant. Sets new state-of-the-art results on ImageNet classification and COCO detection.
- Model-agnostic nature allows wide applicability to various KD methods and models. Simple extension can benefit the community compared to more complex techniques.

**Weaknesses:**

- The paper should address the lack of novelty by acknowledging that feature normalization techniques have already been widely employed in knowledge distillation. For example, PKD (NeurIPS-2023) specifically incorporates channel alignment for detectors, and SKD (Guo Jia) explores normalization techniques on predictions. and Feature Normalized Knowledge Distillation for
/mage Classification ECCV2022 also presents feature norm. Furthermore, it is worth investigating whether the proposed method has already been considered in the distiller's search work, as exemplified by KD-Zero: Evolving Knowledge Distiller for Any Teacher-Student Pairs (NeurIPS-2023).

- In addition, the paper should incorporate a thorough discussion of relevant KD-related studies, including Self-Regulated Feature Learning via Teacher-free Feature Distillation (ECCV2022), NORM: Knowledge Distillation via N-to-One Representation Matching (ICLR2023), Shadow Knowledge Distillation: Bridging Offline and Online Knowledge Transfer (NIPS2022), DisWOT: Student Architecture Search for Distillation Without Training (CVPR2023), and Automated Knowledge Distillation via Monte Carlo Tree Search (ICCV2023). These discussions will provide valuable insights into the existing literature, establish connections with previous research, and potentially highlight points of comparison and contrast.

**Questions:**

The only concern to me is the novelty of the work and I hope the authors could discuss some of the related work I mentioned in the revised version.

---

### Official Review · Reviewer_VRvE · 2023-10-31

**Soundness:** 3 good
**Presentation:** 3 good
**Contribution:** 3 good
**Rating:** 6
**Confidence:** 4

**Summary:**

This paper studies Knowledge Distillation (KD). A simple loss term namely ND loss is proposed to enhance the distillation performance. It encourages the student to produce large-norm features and aligns the direction of student features and teacher class-means. The ND loss helps not only logit-based distillation methods but also feature-based distillation methods.

**Strengths:**

1. The proposed method is simple but effective. Encouraging the feature norm for the student is novel in the field of KD.
2. Experimental results are strong. The authors also conduct experiments on object detection. The proposed loss can improve the existing methods on both image classification and object detection.
3. The whole paper is organized and written well.

**Weaknesses:**

It is not a novel thing that decoupling the feature into the magnitude and the direction. Previous works [1][2] already studied this point. [1] uses the teacher classifier to project both teacher features and student features into the same space and then align them. [2] proposes a loss term to align two features’ direction. Compared to the existing works, this paper proposes enlarging feature norm and utilizing the class-mean feature. Authors should check more existing papers and discuss their differences.
[1] Yang, Jing, et al. "Knowledge distillation via softmax regression representation learning." International Conference on Learning Representations (ICLR), 2021.

[2] Wang, Guo-Hua, Yifan Ge, and Jianxin Wu. "Distilling knowledge by mimicking features." IEEE Transactions on Pattern Analysis and Machine Intelligence 44.11 (2021): 8183-8195.

**Questions:**

None

---

### Official Review · Reviewer_AuzT · 2023-10-31

**Soundness:** 2 fair
**Presentation:** 2 fair
**Contribution:** 2 fair
**Rating:** 5
**Confidence:** 4

**Summary:**

This paper proposes to use teacher's class-mean to align student's direction and encourage the student to produce large-norms features, improving the performance of KD.

**Strengths:**

The paper is generally well-written, and the methodology is well-motivated.

**Weaknesses:**

1. would expect comparisons and discussion to similarity-preserving KD e.g., [1], which is a large family in feature distillation methods and shows some relations to the proposed method.
2. Meanwhile, comparisons/discussion to explainablity-based KD, e.g., [2] are needed to see whether those methods can be benefited from the proposed method.

[1] Tung, Fred, and Greg Mori. “Similarity-Preserving Knowledge Distillation.” ICCV 2019.

[2] Guo, Ziyao, et al. "Class Attention Transfer Based Knowledge Distillation." CVPR 2023.

**Questions:**

Please see weakness.